# Acute Care of Older Patients with COVID-19: Clinical Characteristics and Outcomes

**DOI:** 10.3390/geriatrics5040065

**Published:** 2020-09-27

**Authors:** Zara Steinmeyer, Sara Vienne-Noyes, Marc Bernard, Armand Steinmeyer, Laurent Balardy, Antoine Piau, Sandrine Sourdet

**Affiliations:** Geriatrics, Centre Hospitalier Universitaire de Toulouse, 31059 Toulouse, France; sara.viennenoyes@yahoo.fr (S.V.-N.); bernard.m@chu-toulouse.fr (M.B.); armand.steinmeyer@gmail.com (A.S.); balardy.l@chu-toulouse.fr (L.B.); piau.a@chu-toulouse.fr (A.P.); sourdet.s@chu-toulouse.fr (S.S.)

**Keywords:** older adults, COVID-19, frailty, mortality

## Abstract

(1) Background: COVID-19 has become a global pandemic and older patients present higher mortality rates. However, studies on the characteristics of this population set are limited. The objective of this study is to describe clinical characteristics and outcomes of older patients hospitalized with COVID-19. (2) Methods: This retrospective cohort study was conducted from March to May 2020 and took place in three acute geriatric wards in France. Older patients hospitalized for COVID-19 infections were included. We collected clinical, radiological, and laboratory outcomes. (3) Results: Ninety-four patients were hospitalized and included in the final analysis. Mean age was 85.5 years and 55% were female. Sixty-four (68%) patients were confirmed COVID-19 cases and 30 (32%) were probable. A majority of patients were dependent (77%), 45% were malnourished, and the mean number of comorbidities was high in accordance with the CIRS-G score (12.3 ± 25.6). The leading causes of hospitalization were fever (30%), dyspnea (28%), and geriatric syndromes (falls, delirium, malaise) (18%). Upon follow-up, 32% presented acute respiratory failure and 30% a geriatric complication. Frailty and geriatric characteristics were not correlated with mortality. Acute respiratory failure (*p* = 0.03) and lymphopenia (*p* = 0.02) were significantly associated with mortality. (4) Conclusions: Among older patients hospitalized with COVID-19, clinical presentations were frequently atypical and complications occurred frequently. Frailty and geriatric characteristics were not correlated with mortality.

## 1. Introduction

On 31 December 2019, the World Health Organization (WHO) was alerted about a cluster of pneumonia of unknown etiology in the city of Wuhan, China [1]. This newly emerged Coronavirus, designated Coronavirus 2019 disease (COVID-19) illustrated properties such as its high infectiousness and effects which can cause illness with a range from asymptomatic forms to respiratory failure [2].

On 11 March 2020, COVID-19 was declared a global pandemic [3] and by 15 April 2020 had infected just under 2,000,000 and caused 126,000 deaths, pointing to a major public health crisis [4]. The first COVID-19 cases in France were confirmed in Bordeaux on 24 January 2020.

The first descriptions of epidemiologic characteristics of the COVID-19 outbreak were reported in China. In a February 2020 study, Wu et al. described 72,314 cases with most patients ranging from 30 to 79 years of age and an overall case fatality rate of 2.3% correlated to preexisting comorbidities such as cardiovascular disease, diabetes or chronic respiratory diseases. Fatal cases were primarily concentrated among older patients, in particular those aged ≥80 years with a case fatality of 14.8% [2]. Italy as of 24 April 2020 had a number of fatal cases reaching 26,000 with a case fatality rate of 19.7% in patients aged above 80 [4,5]. Evidence suggests that advanced age is an important predictor for severe forms of COVID-19 infection and mortality [2]. However, old patients are the most heterogenous of any age group and chronological age does not reflect their physiological reserve to overcome acute events [6]. As such, assessing an older patient’s overall health status based on multidisciplinary geriatric assessments is essential in identifying patients most at risk of negative health outcomes [7]. Despite evidence that older adults are more likely to present severe COVID-19 symptoms and higher rates of mortality, there is a lack of data focusing on their clinical presentation [2,8,9,10] and evolution [11].

This study aims to describe clinical characteristics and outcomes in older patients hospitalized with COVID-19 in order to improve patient care. Knowledge of COVID-19 infection in this patient set is a vital component in understanding the clinical evolution of COVID-19 andto determine factors predictive of mortality and acute respiratory failure in a proven at-risk population.

## 2. Materials and Methods

### 2.1. Study Participants

This is a retrospective cohort study on older patients diagnosed with COVID-19 and hospitalized in 3 acute geriatric wards from 13 March to 4 May 2020. Patients were followed from hospital admission to hospital discharge or death.

The patients enrolled in this study were diagnosed according to the WHO interim guidance [12]. Confirmed cases of COVID-19 were defined as patients with laboratory confirmation of the virus: positive reverse-transcription polymerase chain reaction (RT-PCR) test results on samples of nasal swabs irrespective of clinical signs and symptoms. Probable cases of COVID-19 were defined as patients with abnormal chest computed tomography scans (CT-scan) correlated to characteristics of COVID-19 (ground-glass opacity, followed by ground-glass opacity with consolidation, rounded opacities, a crazy-paving pattern, and an air bronchogram), and where other aetiologias explaining the clinical presentation were excluded.

### 2.2. Data Collection

Medical data of patients were obtained from electronic medical records using a standardized data collection form. Information included demographic data, medical history, underlying comorbidities, geriatric assessment, laboratory values, a chest CT-scan upon patient’s arrival in the ward, and clinical outcomes.

Burden of disease in older patients was assessed with the cumulative illness rating scale-geriatric (CIRS-G) [13]. The number of medications was assessed, and polypharmacy was defined as more than 5 medications.

As part of usual care, a geriatric assessment was performed during hospitalization by the physicians in charge of the patient:-Functional abilities were assessed using the Katz Activities of Daily Living (ADL) and Lawton’s Instrumental Activities of Daily Living (IADL) scales upon arrival [14]. Dependency was defined as at least one difficulty in an item.-Frailty status was screened with the Frail Non-Disabled survey (FIND) developed by Césari et al. [15]. Patients were defined accordingly as robust, frail, or dependent. Frail patients are associated in literature with higher risks and negative health outcomes [16].-The nutritional status of the patient was assessed using the Mini Nutritional Assessment (Mini MNA) [17]. Body mass index (BMI) in kg/m² was calculated from measured data.-The Norton scale assessed risk for pressure ulcers. A score <14 was considered a high risk [18].-Social environment assessment included living conditions, marital status, and presence or absence of home health care services.

Routine blood tests were performed, and reference limits were defined by the laboratory of the Toulouse University hospital.

Due to the high rate of severe acute respiratory failure in older patients with COVID-19 [2], advance care planning was discussed upon the patient’s arrival. Physicians assessed the patient’s values and goals of care and discussed preferences for cardiopulmonary resuscitation (CPR) and life-sustaining treatment therapies with patients and/or their surrogate decision makers in the case of severe acute respiratory failure. The possibility to receive CPR or undergo mechanical ventilation was then discussed by physicians (geriatricians and intensive care physicians) in consideration of the patient’s choice, pre-existing chronic life-limiting illnesses, geriatric characteristics, goal of care, and medical effectiveness [19].

Adverse events were recorded during hospitalization:-Acute respiratory failure was defined as an acute episode of supplemental oxygen and signs of respiratory distress observation scales [20]. Respiratory failure type 1 was defined as a low level of oxygen in the blood without an increased level of carbon dioxide [21].-Acute kidney insufficiency was identified according to the kidney disease improving global outcomes definition [22].-Acute heart failure was defined as a rapid onset of new or worsening signs and symptoms of heart failure [23].-Dehydration was identified if patients presented clinical features or biochemical changes of dehydration.-Delirium was identified if patients presented a score above 3 using the confusion assessment method [24].

### 2.3. Statistical Analysis

Descriptive statistics were obtained for all study variables. Continuous data are expressed as mean (SD) or median (interquartile range (IQR)) values. Categorical data is expressed as proportions. Survivors and non-survivors were compared using the Fisher exact test or χ2 test for categorical variables and using the *t*-test or the Mann‒Whitney U test for continuous variables. Univariable and multivariable Cox proportional hazard models with backward selection were performed to identify potential significant prognostic factors of death and acute respiratory failure to determine the hazards ratio (HR) and 95% confidence interval (CI). Variables were entered in the model if they were significant in univariate analysis with a *p*-value < 0.20. Tests were two-sided, and the cut-off significance was set at *p* < 0.05. Data was analyzed using STATA^®^ version 11 (Stata Corp.,College Station, TX, USA).

## 3. Results

### 3.1. Patients Baseline Characteristics

The baseline characteristics are described in Table 1. The mean age of patients was 85.5 years old (range, 62–99), 53 were aged above 85 years old (56.4%) and 52 (55%) were female.

Of the patients hospitalized, 63 (68%) lived at home and 21 (23%) patients lived alone. Concerning patient comorbidities measured with the CIRS-G [13], the total number of categories endorsed was 5.3 ± 2.2 out of 14 organ categories, and the total score was 12.3 ± 5.1 with a severity index of 2.36 ± 0.54. All patients had at least one comorbidity and 84 (89%) of them had at least one severe comorbidity (grade 3 or 4).

The most prevalent severe comorbidities were psychiatric (including dementia) (37 (40%)), and cardiac diseases (23 (25%)). The mean number of medication of patients was 6.3 ± 3.3, and 16 patients (17%) had more than ten medications.

According to the FiND criteria [15], 12 (13%) were robust, 10 (11%) were frail, and 72 (77%) were dependent. The median ADL score of patients was 4 (2–6) and the IADL score was 2 (0–6.5) [14]. Among them, 44 (45%) were malnourished according to the MNA [17] or BMI. Advance care planning was discussed for 90 (96%) patients. Of the patients, life sustaining treatment was recommended for 28 patients (31%) based on the clinical assessment. For baseline characteristics of hospitalized older patients with COVID-19 stratified by age, see the Appendix A.

Table 2 illustrates the admission characteristics of COVID-19 infections in older patients. The leading causes of hospitalization were fever (28 (30%)), dyspnea (26 (28%)), and geriatric syndromes (falls, delirium, malaise) (17 (18%)). Upon arrival, the majority of patients (72 (77%)) presented respiratory tract symptoms. Geriatric syndromes (falls, delirium, malaise) were reported in a total of 45 (48%) of the patients and were exclusive in 5 patients.

The population set under study was composed of 64 (68%) confirmed cases and 30 (32%) probable cases. Ten patients (10.6%) had CT-scans supporting evidence of a COVID-19 infection but did not present any respiratory tract symptoms upon arrival. Among them, the leading causes of hospitalizations were fever (5 (50%)), delirium (2 (20%)), fall (1 (10%)) and fatigue (2 (20%)). Onset of symptom to hospital admission was of 3 days (range from1 to 7 days).

Vital signs were measured on arrival with a mean of pulse was 81.3 ± 22.1 per minute, respiratory rate was 22.8 ± 6.5 per minute, temperature was 37.3 ± 1.0 °C, and systolic blood pressure was 133 ± 25.6. Forty-two (45%) patients required oxygen upon arrival and the mean saturation of patients without oxygen at arrival was 94.5% ± 2.4%.

Table 3 reports the results of complementary exams. Among laboratory measures on admission, anemia was present in 40 (42%) patients and mean hemoglobin count was 12.5 ± 1.8. Sixty-nine (83.1%) patients had high levels of NT-pro-*BNP* with a median of 900 (418–2331) and of hypersensitive troponin I with a median of 33 (23–48).

### 3.2. Patient Follow-Up

Table 4 describes patient’s outcomes during hospitalization. Mean length of hospital stay was 12.0 ± 5.5 days (range from 2 to 31 days). During hospitalization, 78 (83%) patients presented a respiratory failure type 1 and 30 (32%) an acute respiratory failure. Mean onset of symptom to acute respiratory failure was 8.9 ± 5.3 days.

Among deceased patients (*n* = 17), 13 patients presented an acute respiratory failure prior to death and 4 patients died for other reasons (fall, alteration of the general state, and cardiogenic shock). Seven patients (7%) were transferred to an intensive care unit (ICU). During hospitalization, a total of 28 (30%) patients presented a geriatric complication and 81 (87%) patients presented at least one non geriatric complication. The mean ADL change at the end of hospitalization was −0.6+/−1.6 (*n* = 71).

Treatment implemented during hospitalization was composed in the majority by antibiotics (77 (82%)) and venous fluids (54 (58%)). Seven patients (7%) were treated using corticosteroids, 8 (8.6%) had a treatment using hydroxychloroquine, and only one patient (1%) was treated with an antiviral (Remdesivir) as part of a research protocol.

Out of all the patients, 17 died (18%), with 3 in ICU. Thirty-seven were discharged (39%) (25 returned home, 12 returned to their respective nursing homes). Thirty-five (37%) of them were sent to a rehabilitation center. Five (6%) patients are still hospitalized and two patients are in the ICU. The median time of onset of symptom to death was 13 days (range from8 to15 days).

### 3.3. Prognostic Factors of Death and Acute Respiratory Failure

Using univariate Cox regression, age above 85 was shown to increase the likelihood of death, lymphopenia (<0.8G/L), and acute respiratory failure. No other socio-demographic, clinical, biological, or medical imaging variables were significant. In the final model, only acute respiratory failure (adjusted HR = 3.6, CI 95% 1.1–11.4, *p* = 0.03) and lymphopenia (adjusted HR = 4.9, CI 95% 1.3–18.5, *p* = 0.02) were significantly associated with death. For detailed information on multivariate Cox regression for prognostic factors and after adjustment for gender, see the Appendix A. 

Age (adjusted HR = 2.9, CI 95% 1.2–7, *p* = 0.02), functional ability with IADL (adjusted HR = 0.4, CI 95% 0.19–0.97), *p* = 0.04), CRP level (adjusted HR = 2.8, CI 95% 1.3–5.9, *p* < 0.01), and platelet count (adjusted HR = 2.8, CI 95% 1.2–6.6, *p* = 0.02) were significantly associated with acute respiratory failure during hospitalization.

## 4. Discussion

This study describes characteristics and follow-up of 94 older patients (mean age 85.4+/−7.4) hospitalized for a COVID-19 infection. The leading causes of hospitalization were fever (28 (30%)), dyspnea (26 (28%)) and more than 45 (48%) patients presented a geriatric syndrome upon arrival (falls, delirium).

Patients had a high comorbidity burden according to the CIRS-G (12.3 ± 25.6) while 72 (77%) patients were dependent and 42 (45%) malnourished. Among them, 7 (7%) patients were transferred to an ICU, 37 (39%) patients were discharged, and 35 (37%) patients went to a rehabilitation center. The overall case fatality rate was 18%.

Among the patient set, 18% were admitted for signs or symptoms (falls, malaise, or delirium) unrelated to typical COVID-19 presentations. Such a prevalence of atypical symptoms in this study confirms the great variety of presentations of the infection in this age group, confirming studies that have reported COVID-19 diagnosis after admission for syncope and delirium [25], or fall in older adults [26]. Further, an anomalous clinical presentation of COVID-19 heightens the risk of misdiagnosis leading to inappropriate medical care without quarantine and can be particularly dangerous in terms of infectiousness when considering nursing homes [18,27].

Diagnosis of older patients in this study was further complicated due to a number of false negative RT-PCR results (32%), comparable to other studies [28]. This in addition to atypical presentation are two factors that could misguide diagnosis in older patients. Positive RT-PCR nasopharyngeal swabs produced a rate of confirmed cases using laboratory measures at 68% in older patients, albeit with a smaller population set, in comparison with Wu et al.’s 62% rate of confirmed cases in China [2]. Nevertheless, CT-scans have proven to be effective in diagnosing patients with COVID-19 disease by providing an additional variable for physicians and should be more systematically proposed in the case of negative PCR, during the pandemic period. In our study, CT-scans allowed for the diagnosis of 30 (32%) patients with negative PCR, 6 (6%) of them with no respiratory symptoms at admission. However, in this population CT-scan interpretation remains complex due to underlying comorbidities (acute heart failure, chronic respiratory disease).

In terms of clinical outcomes, a large number of patients developed non geriatric (87%) and geriatric complications (30%) during hospitalization. Prevalence of acute respiratory failure was higher than in other studies (32%) [29,30]. Fall (12%) and delirium (13%) were the most frequent geriatric complications during hospitalization. The high occurrence of these complications was not surprising given the prevalence of comorbidities and frailty in our population. These complications must be taken into account by physicians, and targeted preventive measures should be implemented in this population set.

Although older age is an important predictor of mortality in COVID-19 patients in the general population, age itself does not seem to be predictive above a certain threshold [2]. Case mortality rates of this study’s patient set stood at 18% for a patient mean age of 85.5 and is aligned with those found in Italy of 20% for a mean age of 79.5 [31] and China (15%) without a defined mean age [2].

This study observed that comorbidities, polypathology, and in particular chronic respiratory, cardiovascular or diabetes were not correlated with mortality in this population set as opposed with other studies [2]. The high prevalence of comorbidities and the sample size of this study may explain these results.

Respiratory failure during hospitalization (*p* = 0.03) and lymphopenia upon hospital admission (*p* = 0.02) were significantly correlated with mortality. Death in COVID-19 infections is primarily associated to the onset of an acute respiratory failure, and in our study 4 patients died of other causes (e.g., fall, alteration of the general state…) [10]. Moreover, lymphopenia was associated with mortality in our study, as reported by previous authors there is evidence that it may predict disease severity and prognosis in COVID-19 patients [32].

The likelihood of acute respiratory failure was associated with a number of factors that were measured in the multivariate analysis. Higher CRP levels and platelet count in the event of acute respiratory failurewas identified and is aligned with the literature on the role of major inflammation as a cause for poor outcomes in older patients [33]. On the other hand, functional ability with IADL seemed to have a protective effect, which is in line with our hypothesis given that patients with greater functional ability have a better health status. Nevertheless, other geriatric characteristics such as nutrition and frailty status did not have any bearing on the outcome. De smet et al. show in a similar population that COVID-19 mortality was associated with frailty status and viral load; other geriatric characteristics such as long-term care residence, cognition, and medication were not associated with mortality [34]. Further research is needed to discern which factors associated with better functional ability explain its protective effect.

Geriatric assessments may not predict death due to COVID-19, at this point, but could help identify patients who are at high risk of developing long-term complications as well as complications due to ICU and those who will benefit from an ICU [35]. As such, it remains essential to take into account those parameters while taking care of COVID-19 older patients and when discussing advance care planning. This study noted that goal-concordant care between the patient and physicians is essential to avoid intensive life-sustaining treatments when they are non-beneficial, unwanted by patients, and if health care capacity and resources are limited [19].

### Limitations

This study has some limitations. The main limitation is the small sample size limiting data interpretation and missing data concerning survival analysis. Secondly, the patient set was drawn from three acute geriatric wards in Toulouse University Hospital creating a population bias. Third, the follow-up time is relatively short to evaluate potential medium- and long-term consequences.

## 5. Conclusions

In this case series, a majority of older patients hospitalized with COVID-19 were dependent and comorbid. Frailty and geriatric characteristics were not correlated with mortality in this population set. These results may provide valuable insight for the care of elderly patients affected by COVID-19 and may play an important role when discussing advance care planning.

## Figures and Tables

**Table 1 geriatrics-05-00065-t001:** Baseline characteristics of older patients with COVID-19.

Characteristics (*n* = 94)	Mean ± SD or N (%)	Survivors (*n* = 77)	Non-Survivors (*n* = 17)	*p*
**Socio-Demographic**				
**Age**	85.5 ± 7.5	84.8 ± 7.8	88.6 ± 5.3	0.06
**Gender**, Female	52 (55.3)	41 (53.2)	11 (64.7)	0.39
**Marital status** (*n* = 93) Single Married Widowed Divorced	8 (8.6) 42 (45.2) 35 (37.6) 8 (8.6)	6 (7.9) 35 (46.0) 28 (36.8) 7 (9.2)	2 (11.8) 7 (41.2) 7 (41.2) 1 (5.9)	0.90
**Living Location** Private household Assisted living Nursing home Other	63 (68.1) 7 (7.4) 22 (23.4) 1 (1.1)	53 (68.8) 6 (7.8) 17 (22.1) 1 (1.3)	11 (64.7) 1 (5.9) 5 (29.4) 0	0.89
**Home health care services** Nurse Home health aides	58 (61.7) 58 (61.7)	50 (64.9) 51 (66.2)	8 (47.1) 7 (41.2)	0.17 0.05 *
**Health conditions**				
**Cumulative Illness Rating Scale (CIRS-G)**				
Total score	12.3 ± 5.1	12.1 ± 5.2	13 ± 4.9	0.55
Cardiac disease	46 (48.9)	35 (45.4)	11 (64.7)	0.15
Hypertension	65 (69.1)	52 (67.5)	13 (76.5)	0.57
Respiratory disease	35 (37.2)	30 (39.0)	5 (29.4)	0.46
Diabetes mellitus	11 (11.7)	9 (11.7)	2 (11.8)	0.99
Dementia	43 (45.7)	34 (44.2)	9 (52.9)	0.51
**Medication**				
Polypharmacy (Drugs ≥5)	65 (69.1)	51 (66.2)	14 (82.3)	0.19
Presence of ACEI or ARA	33 (35.1)	24 (31.2)	9 (52.9)	0.09
Presence of antiplatelet/anticoagulant therapy	49 (52.1)	38 (49.3)	11 (64.7)	0.25
**Functional assessment**				
**FIND** Dependant Frail Robust	72 (76.6) 10 (10.6) 12 (12.8)	60 (77.9) 8 (10.4) 9 (11.7)	12 (70.6) 2 (11.8) 3 (17.6)	0.78
**ADL score (/6),** ≥1 dependency	61 (64.9)	50 (64.9)	11 (64.7)	0.99
**IADL score (/8),** ≥1 dependency	70 (76.1)	58 (77.3)	12 (70.6)	0.56
**Nutritional assessment**				
**Mini MNA** Normal nutritional status (12–14) Risk of malnutrition (8–11) Malnutrition (≤7)	17 (18.1) 35 (37.2) 42 (44.7)	14 (18.2) 30 (39.0) 33 (42.9)	3 (17.6) 5 (29.4) 9 (52.9)	0.78
**BMI (kg/m2)** (*n* = 92) Underweight (BMI <21) Normal weight (BMI 18–25) Overweight (BMI 25–30) Obese (BMI >30)	17 (18.5) 33 (35.9) 29 (31.5) 13 (14.1)	14 (18.4) 27 (35.5) 24 (35.5) 11 (14.5)	3 (18.7) 6 (37.5) 5 (31.2) 2 (12.5)	0.99
**Norton scale** (*n* = 91) <14 (Pressure ulcer risk)	32 (35.2)	26 (34.7)	6 (37.5)	0.83

Abbreviations: SD: Standard deviation; IQR: interquartile; ACEI: angiotensin-converting enzyme inhibitor, ARA: angiotensin II receptor antagonists, CIRS-G: Cumulative Illness Rating Scale Geriatric; FiND: Frail Non-Disabled questionnaire; ADL: Activities of Daily Living; IADL: Instrumental Activities of Daily Living; Mini MNA: Mini Nutritional Assessment; BMI: body mass index; * *p* < 0.05.

**Table 2 geriatrics-05-00065-t002:** Admission characteristics of older patients with COVID-19.

Characteristics (*n* = 94)	Mean ± SD or N (%)	Survivors (*n* = 77)	Non-Survivors (*n* = 17)	*p*
**COVID-19 Cases**				
Confirmed cases: RT-PCR (+)	64 (68.1)	51 (66.2)	13 (76.5)	0.41
Probable cases: RT-PCR (−) / CT-scan (+)	30 (31.9)	26 (33.7)	4 (23.5)	0.57
**Vital Signs Upon Arrival**				
Pulse >100 per min	17 (18.1)	14 (18.2)	3 (17.6)	0.96
Respiratory rate (*n* = 90) >30 breaths per min	7 (7.8)	4 (5.5)	3 (17.6)	0.09
Fever (Temperature ≥38 °C)	24 (25.5)	20 (26.0)	4 (23.5)	0.99
Systolic blood pressure <90 mmHg	1 (1.1)	0	1 (5.9)	0.99
**Symptoms and Signs**				
**Leading cause for patient admission (*n* = 93)** Cough Dyspnea Chest pain Fever Malaise Abdominal pain Delirium Fall Fatigue	8 (8.6) 26 (28.0) 1 (1.1) 28 (30.1) 1 (1.1) 1 (1.1) 8 (8.6) 8 (8.6) 12 (12.9)	4 (5.3) 23 (30.3) 1 (1.3) 23 (30.3) 1 (1.3) 1 (1.3) 8 (10.5) 6 (7.9) 9 (11.8)	4 (23.5) 3 (17.6) 0 5 (29.4) 1 (1.3) 1 (1.3) 0 2 (11.8) 3 (17.6)	0.4
**Admission Symptoms**				
Runny nose	4 (4.3)	4 (5.2)	0	0.34
Pharyngalgia	2 (2.1)	2 (2.6)	0	0.50
Anosmia	3 (3.2)	3 (3.9)	0	0.41
Dry cough	36 (38.3)	30 (39.0)	6 (35.3)	0.78
Sputum production	18 (19.1)	16 (20.8)	2 (11.8)	0.39
Dyspnea	52 (55.3)	41 (53.2)	11 (64.7)	0.39
Chest pain	4 (4.3)	4 (5.2)	0	0.34
Fever	56 (59.6)	46 (59.7)	10 (58.8)	0.94
Chills	10 (10.6)	10 (13.0)	0	0.12
Malaise	7 (7.4)	7 (9.1)	0	0.20
Headache	4 (4.3)	3 (3.9)	1 (5.9)	0.71
Abdominal pain	9 (9.6)	6 (7.8)	3 (17.6)	0.21
Diarrhea	12 (12.8)	10 (13.0)	2 (11.8)	0.89
Muscle ache	10 (10.6)	8 (10.39)	2 (11.8)	0.87
Delirium	30 (31.2)	27 (35.1)	3 (17.6)	0.16
Fall	21 (22.6)	16 (20.8)	5 (29.4)	0.44
Fatigue	64 (68.1)	50 (64.9)	14 (82.3)	0.16
Vomiting	8 (8.5)	5 (6.5)	3 (17.6)	0.14

Abbreviations: SD: standard deviation; RT-PCR: reverse transcriptase polymerase chain reaction; CT-scan: computed tomography scan.

**Table 3 geriatrics-05-00065-t003:** Complementary exams of older patients with COVID-19.

Characteristics (*n* = 94)	Mean ± SD or N (%)	Survivors (*n* = 77)	Non-Survivors (*n* = 17)	*p*
**Admission Laboratory Measures**				
Anemia _(a)_	40 (42.5)	34 (44.2)	6 (35.3)	0.50
White Blood cell count, × 10^9^/L <4 4–10 >10	11 (11.7) 63 (67.0) 20 (21.3)	11 (14.3) 50 (64.9) 16 (20.8)	0 13 (76.5) 4 (23.5)	0.25
Neutrophil count, × 10^9^/L (*n* = 86) <2 2–7.5 >7.5	7 (8.1) 55 (63.9) 24 (27.9)	7 (9.1) 47 (61.0) 23 (29.9)	8 (47.1) 0 9 (52.9)	0.12
Absolute lymphocyte count, × 10^9^/L (*n* = 87) <0.8	24 (27.9)	17 (23.0)	7 (58.3)	0.01 *
Platelet count, × 10^9^/L <150 150–450 <450	19 (20.2) 72 (76.6) 3 (3.2)	14 (18.2) 60 (77.9) 3 (3.9)	5 (29.4) 12 (70.6) 0	0.44
Sodium, mmol/L <136 136–145 >145	33 (35.1) 53 (56.4) 8 (8.5)	26 (33.8) 44 (57.1) 7 (9.1)	7 (41.2) 9 (52.9) 1 (5.9)	0.81
Potassium, mmol/L <3.4 3.4–4.5 >4.5	7 (7.4) 71 (75.5) 16 (17.0)	4 (5.2) 61 (79.2) 12 (15.6)	3 (17.6) 10 (58.8) 4 (23.5)	0.12
Blood urea nitrogen, mmol/L >8.21	40 (43.0)	32 (42.1)	8 (47.1)	0.71
Creatinine, median (IQR), μmol/L	89 (70–120)	89 (70–115)	107 (70-140)	0.61
Aspartate aminotransferase, UI/L (*n* = 91) >35	44 (48.3)	33 (44.6)	11 (64.7)	0.13
Alanine aminotransferase, UI/L *(n* = 91) >35	37 (40.1)	30 (40.5)	7 (41.2)	0.96
Gamma-glutamyl transferase UI/L (*n* = 91) >40	38 (41.7)	30 (40.5)	8 (47.1)	0.62
Alkaline phosphatase UI/l (*n* = 90) >104	19 (21.1)	13 (17.8)	6 (35.3)	0.11
C-reactive protein mg/l	55 (21–107)	50 (16–99)	85 (51–124)	0.11
NT-pro-*BNP* pg/mL (*n* = 83) >300	69 (83.1)	54 (79.4)	15 (100)	0.06
Hypersensitive troponin I, ng/mL (*n* = 70) <14	48 (68.6)	38 (65.5)	10 (83.3)	0.23
Albumin g/l (*n* = 86) <30	50 (58.1)	40 (54.8)	10 (76.9)	0.14
pH (*n* = 75) <7.35 7.35–7.45 >7.45	3 (4.0) 36 (48.0) 36 (48.0)	3 (5) 30 (50) 27 (45)	0 6 (40) 9 (60)	0.69
Arterial blood gas				
P02, mmHg (*n* = 75) <70	38 (50.7)	29 (48.3)	9 (60)	0.56
PC02, mmHg (*n* = 75) <35 35–48 >48	48 (64) 26 (34.7) 1 (1.3)	38 (63.3) 22 (36.7) 0	10 (66.7) 4 (26.7) 1 (6.7)	0.21
Bicarbonate mmol/L (*n* = 76) <22 22–29 >29	35 (46.0) 38 (50) 3 (3.9)	29 (47.5) 30 (49.2) 2 (3.3)	6 (40) 8 (53.3) 1 (6.7)	0.60
Lactate, mmol/L (*n* = 74) >2.2	11 (14.9)	10 (16.9)	1 (6.7)	0.32
**Chest CT-Scan Findings in Favor of COVID 19 (n = 86)**				
Pneumonia				
Unilateral	8 (9.3)	6 (8.6)	2 (12.5)	0.64
Bilateral	50 (58.1)	39 (55.7)	11 (68.7)	0.41
Ground glass opacities	74 (86.0)	59 (84.3)	15 (93.7)	0.45
Pleural effusion	20 (23.3)	16 (22.9)	4 (25)	0.99

Abreviations: SD: standard deviation; IQR: interquartile; RT-PCR: reverse transcriptase polymerase chain reaction; CT-scan: computed tomography thoracic scan; NT-pro-BNP: N-terminal-pro-BNP; Pa02: partial pressure of oxygen; PaC02 partial pressure of Carbone dioxide. SI conversion factors: to convert alanine amino transferase to μkat/L, multiply by 0.0167; aspartate amino transferase to μkat/L, multiply by 0.0167; gamma glutamyl transferase to μkat/L, alkaline phosphatase to μkat/L, multiply by 0.0167; BNP to ng/L, multiply by 1; troponin I to μg/L, multiply by 1; oxygen P02 to kPa, multiply by 0.133; carbon dioxide, PC02 to kPa, multiply by 0.133. _(a)_ Anemia: hemoglobin count <12g/L for women and <13g/L for men according to WHO. * *p* < 0.05.

**Table 4 geriatrics-05-00065-t004:** Follow-up and outcomes of older patients with COVID-19 infection.

Characteristics (*n* = 94)	Mean ± SD or N (%)	Survivors (*n* = 77)	Non-Survivors (*n* = 17)	*p*
**Outcomes during hospitalization**				
**Geriatric complications**				
Delirium during hospitalization	11 (11.7)	7 (9.1)	4 (23.5)	0.11
Fall	12 (12.7)	11 (14.3)	1 (5.9)	0.69
Pressure ulcer	5 (5.3)	4 (5.2)	1 (5.9)	0.99
Fecal impaction (*n* = 92)	4 (4.4)	3 (3.9)	1 (6.2)	0.54
**Non geriatric complications**				
Acute respiratory failure	30 (31.9)	17 (22.1)	13 (76.5)	0.00 *
Onset of symptoms to acute respiratory failure (days)	8.9 ± 5.3	9.8 ± 5.8	7.7 ± 4.4	0.29
Respiratory failure type 1	78 (83)	61 (79.2)	17 (100)	0.04
Thrombosis or pulmonary embolism	3 (3.2)	1 (1.3)	2 (11.8)	0.08 *
Acute heart failure (*n* = 92)	22 (23.7)	15 (19.7)	7 (41.2)	0.11
Acute kidney failure	32 (34.0)	26 (33.8)	6 (35.3)	0.99
Pain	20 (21.3)	14 (18.2)	6 (35.3)	0.19
Dehydration	34 (36.2)	29 (37.7)	5 (29.4)	0.59
Skin manifestations	11 (11.7)	9 (11.7)	2 (11.8)	0.99

Abbreviation: SD: standard deviation; * *p* < 0.05.

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
