# Peer review of "Acute Care of Older Patients with COVID-19: Clinical Characteristics and Outcomes"

_geriatrics, 2020, doi:10.3390/geriatrics5040065_

Round 1
Reviewer 1 Report
With this study, the authors examined in a small group of older persons the risk factors associated with higher mortality during COVID-19 infection. The authors concluded that “among the oldest old, COVID-19 clinical presentations were frequently atypical”-which is already well known- and that “ frailty and geriatric status were not correlated with mortality”. The paper would be of interest, but I have several concerns about its publication. I would like to reconsider it after a major revision.
- The title is wrong. Here the authors are presenting a group of older persons (age range, 62-99), not only oldest one.
- The introduction does not really guide the reader to the main hypothesis and the work novelty.
- The reader is somewhat lost right from the start because of the lack of a guiding framework that helps with the evaluation of the information that is presented.
- The sample size looks really small. Please calculate the exact power. If possible, please include more cases to support findings and conclusion.
- Baseline characteristics of older patients with COVID-19 stratified by age range would be of great interest.
- It seems that there is no difference between survivors or not in the average age. Thus results do not support the conclusions.
- As far as the regression analysis is concerned to test whether age above 85 is associated with higher mortality it should be controlled at least for sex and the presence of ACEI or ARA use. I think with a multivariate the association is not significant.
- It is not clear the classification in dependent, frail and robust. Strange that persons classified as fit have more probability to die. There is some confounding variable to take into account (sex?). Or maybe is the small simple size. An analysis by sex would be of interest. This point is really important, the crucial one, and if the finding is confirmed a big discussion with available literature needs to be reported.
- I don’t understand what means geriatric status..
- From table 3 it seems that all variables indicating a malnutrition status in theelderly are lower in non survivor. I would include a nutritional status z score for analysis.
- Discussion needs a major improvement. It should follow the results and guide the reader to the main finding. Here is really confusing.
- There are many structural errors. The manuscript’ structure should definitely be improved.
Reviewer 2 Report
This is a nice piece in an area which has been neglected. It is straightforward, easy to understand and follow. The writing is clear and I have no problem with the English.
I have a couple of suggestions that might improve the m/s.
- It would be good to show the risk factors for severe disease and mortality in the form of a forest plot. This makes is easy to see how strong an influence the particular variable has either on mortality, but more importantly in this paper on the development of respiratory failure and the need for ITU.
- I think the work would be improved with some context. Could the authors add some data in the discussion on what proportion of a similar population , with similar age, sex and comorbidity profile, would be likely to die in a hospital setting? Just to compare them to this group.
- I think that the authors are somewhat underplaying the results. It is important that many (most) of these patients are not presenting with typical covid symptoms. The presentations are highly variable and consist of a wide range of symptoms and problems. I think that the message is that during a pandemic we cannot be sure that any elderly patient does not have covid. This point could be made more clearly.
A clear nice piece of work.
Round 2
Reviewer 1 Report
Please:
1) Point 3: Also for descriptive analyses a power calculation it is important. Please may you be so kind to calculate it?
1) Point 4: I still would like to see a descriptive tables as Table 1 but stratified in >85 and over 85 with relative statistics.
1) Point 5: I would like to see a final model as indicated before also including sex.
